# Contributions of Circulating microRNAs for Early Detection of Lung Cancer

**DOI:** 10.3390/cancers14174221

**Published:** 2022-08-30

**Authors:** Jody Vykoukal, Johannes F. Fahrmann, Nikul Patel, Masayoshi Shimizu, Edwin J. Ostrin, Jennifer B. Dennison, Cristina Ivan, Gary E. Goodman, Mark D. Thornquist, Matt J. Barnett, Ziding Feng, George A. Calin, Samir M. Hanash

**Affiliations:** 1Department of Clinical Cancer Prevention, The University of Texas MD Anderson Cancer Center, 1515 Holcombe Boulevard, Houston, TX 77030, USA; 2McCombs Institute for the Early Detection and Treatment of Cancer, The University of Texas MD Anderson Cancer Center, Houston, TX 77030, USA; 3Department of Translational Molecular Pathology, The University of Texas MD Anderson Cancer Center, Houston, TX 77030, USA; 4Department of General Internal Medicine, The University of Texas MD Anderson Cancer Center, Houston, TX 77030, USA; 5Department of Experimental Therapeutics, The University of Texas MD Anderson Cancer Center, Houston, TX 77030, USA; 6Fred Hutchinson Cancer Research Center, Seattle, WA 98109, USA

**Keywords:** lung cancer, protein biomarkers, microRNA, early detection

## Abstract

**Simple Summary:**

Blood-based cancer biomarkers are a minimally invasive means to achieve improved assessment of risk and/or earlier detection of lung cancer. Using serum samples from individuals that went on to develop lung cancer within one year following blood draw and from matched controls, we investigated 30 microRNAs for individual and combined utility to discriminate lung cancer cases from controls. We additionally assessed the contributions of top-performing microRNA candidates for improving on the performance of a previously validated four-protein marker panel for lung cancer detection. The results of this study indicate complementary performance of combining circulating microRNA and the four-protein-marker panel in assays for early detection of lung cancer.

**Abstract:**

There is unmet need to develop circulating biomarkers that would enable earlier interception of lung cancer when more effective treatment options are available. Here, a set of 30 miRNAs, selected from a review of the published literature were assessed for their predictive performance in identifying lung cancer cases in the pre-diagnostic setting. The 30 miRNAs were assayed using sera collected from 102 individuals diagnosed with lung cancer within one year following blood draw and 212 controls matched for age, sex, and smoking status. The additive performance of top-performing miRNA candidates in combination with a previously validated four-protein marker panel (4MP) consisting of the precursor form of surfactant protein B (Pro-SFTPB), cancer antigen 125 (CA125), carcinoembryonic antigen (CEA) and cytokeratin-19 fragment (CYFRA21-1) was additionally assessed. Of the 30 miRNAs evaluated, five (miR-320a-3p, miR-210-3p, miR-92a-3p, miR-21-5p, and miR-140-3p) were statistically significantly (Wilcoxon rank sum test *p* < 0.05) elevated in case sera compared to controls, with individual AUCs ranging from 0.57–0.62. Compared to the 4MP alone, the combination of 3-miRNAs + 4MP improved sensitivity at 95% specificity by 19.1% ((95% CI of difference 0.0–28.6); two-sided *p*: 0.006). Our findings demonstrate utility for miRNAs for early detection of lung cancer in combination with a four-protein marker panel.

## 1. Introduction

Lung cancer is the leading (18%) cause of cancer mortality worldwide, accounting for an estimated 1.8 million deaths [1]. Five-year relative survival rates among lung cancer subtypes are closely linked with tumor stage at time of diagnosis, with an estimated overall survival rate of 59.8% when diagnosis is made at the localized stage, 32.9% at the regional stage, and 6.3% at the distant stage, according to recent United States National Cancer Institute (NCI) Surveillance, Epidemiology, and End Results (SEER) Program data [2]. NCI National Lung Cancer Screening Trial (NLST) data indicate that a 20% reduction in lung cancer mortality is achievable in high-risk populations through individual screening with low-dose computed tomography (LDCT) [3]. Similar findings have since been reported based on results from the Dutch–Belgian Nederlands–Leuvens Longkanker Screenings Onderzoek (NELSON) lung cancer screening trial [4]. Nevertheless, translation of LDCT to the general population has proven challenging, and the implementation of imaging-based lung cancer screening to reduce lung cancer disease burden is likely to remain low in the near term. Liquid biopsies offer an ideal minimally invasive means of individualized risk assessment to address the need to improve lung cancer screening, either by identifying subjects at increased risk of lung cancer that would benefit from screening or for complementing LDCT-based screening, notably for assessment of indeterminate imaging findings [5].

MicroRNAs (miRNAs) are ~20–22-nucleotide non-coding RNAs that participate in regulation of gene expression, typically by binding to target miRNA transcripts and thus repressing miRNA translation. Dysregulation of the miRNA landscape was observed in multiple disease states, including cancer [6]. In lung cancers, altered miRNA expression has been found and characterized in tumor tissues as well as body fluids, suggesting utility for miRNAs as potential biomarkers in minimally invasive, liquid biopsy assays for assessing risk or detecting and monitoring disease status [7].

Here, using a cohort of sera (Table 1) collected up to one year prior to diagnosis of lung cancer (*N* = 102) and controls matched for age, sex, and smoking status (*N* = 212), we validated a panel of 30 miRNAs (Table 2) for predictive performance in distinguishing lung cancer cases from controls. The panel was developed based on review of previous reports of differential expression in lung cancer clinical specimens [8,9,10,11,12,13]. We further assessed the contributions of miRNAs for improving upon the predictive performance of a previously validated four-protein marker panel (4MP) consisting of cancer antigen 125 (CA125), carcinoembryonic antigen (CEA), cytokeratin-19 fragment (CYFRA 21-1), and the precursor form of surfactant protein B (Pro-SFTPB) for distinguishing lung cancer cases from controls compared to the 4MP alone [14,15].

## 2. Materials and Methods

### 2.1. Study Population

Serum samples for this nested case–control validation study were selected from the Carotene and Retinol Efficacy Trial (CARET) [139]. CARET was a randomized, double-blind, placebo-controlled trial conducted to assess the safety and cancer prevention efficacy of daily supplementation with β-carotene plus retinyl palmitate in 18,314 persons at high risk for lung cancer. Participants were enrolled at six United States study centers from 1985 to 1994 and were followed for cancer and mortality outcomes through 2005. Cancer status was confirmed based on review of clinical records and pathology reports. All CARET participants provided informed consent at recruitment and throughout follow-up, and the study was conducted with oversight from the respective institutional review boards at each of the study centers. CARET pre-diagnostic serum samples collected up to one year prior to diagnosis from 102 ever-smoker patients with lung cancer and 212 smoking-status-, age-, and sex-matched controls (Table 1) were used to test the individual performance of a panel of 30 candidate miRNAs (Table 2). Samples used for miRNA testing were aliquots of CARET specimens previously used for developing a four protein biomarker panel (4MP) for assessing lung cancer risk based on cancer antigen 125 (CA125), carcinoembryonic antigen (CEA), cytokeratin-19 fragment (CYFRA 21-1), and the precursor form of surfactant protein B (Pro-SFTPB) [14,15].

### 2.2. miRNA Isolation

Expression analysis of 30 total micro RNAs were performed using validation cohort of pre-diagnostic CARET serum samples (*N* = 102) collected up to one year prior to diagnosis of lung cancer and corresponding controls (*N* = 212) matched for age, sex, and smoking status. Total RNAs were isolated from 200 µL of serum using Norgen silicon-carbide resin Total RNA Purification Kit 17250 (Norgen Biotek, Thorold, ON, Canada) and eluted with 25 µL of elution solution. For normalization of sample-to-sample variation in the RNA isolation step, 10 fmol of *Caenorhabditis elegans* cel-miR-39 and cel-miR-54 (mirVana miRNA mimic, Applied Biosystems, Foster City, CA, USA); in a total volume of 5 µL were added to each denatured sample after mixing the serum sample with lysis buffer. Total RNA concentrations were quantified in each sample via a NanoDrop ND-1000 spectrophotometer.

### 2.3. TaqMan Assay

RNA was reverse transcribed using the TaqMan miRNA Reverse Kit (Applied Biosystems) in a 15 μL multiplex RT reaction. Relative quantification of targets was performed by qRT-PCR in reaction buffer containing SYBR Green dye (SsoAdvanced Universal SYBR SuperMix 1725270, Bio-Rad Laboratories, Hercules, CA, USA). The intercalation of SYBR Green into the PCR products was monitored in real time using a CFX384 Touch Real-time PCR Detection System (Bio-Rad Laboratories). Ct values were normalized to miR-16-5p.

### 2.4. Statistical Analyses

Receiver operating characteristic (ROC) curve analysis was performed to evaluate the predicative performance of biomarkers for distinguishing lung cancer cases from matched controls. A combination rule for miRNA was developed using logistic regression models. The combined model of the miRNA panel plus the protein 4MP was developed by fitting a logistic regression with the miRNA panel score and the 4MP scores as two separate predictors. The estimated AUC of the proposed 3-marker miRNA panel and miRNA panel plus 4MP were derived by using the empirical ROC estimator of the linear combination corresponding to the respective models. One thousand bootstrap iterations with replacement were implemented to estimate the confidence intervals and *p* values. Statistical significance was considered at *p* < 0.05.

## 3. Results

### 3.1. Candidate miRNA Selection

A panel of 30 candidate miRNAs was developed for testing based on review of literature reporting of differential expression in lung cancer patient biofluids or tumor tissues relative to control specimens. Priority was given to candidates identified across multiple independent studies. Table 2 summarizes the test panel candidates, with observed up- or downregulation relative to controls, and the number of studies in lung cancer that meet the criteria of: (i) inclusion of clinical samples and (ii) relevant controls, (iii) a validation component, and (iv) significance *p* ≤ 0.05 for the indicated miRNA. Relevant references for previous reports of dysregulated miRNA expression in clinical samples are also summarized in Table 2.

### 3.2. Predictive Performance of miRNAs for Early Detection of Lung Cancer

Assay of the panel of 30 miRNAs was performed using 102 case serum samples collected one-year preceding diagnosis of lung cancer and 212 control serum samples matched on the basis of age, sex, and smoking history (Table 1).

Four of the 30 miRNAs were below the limit of quantitation. Five miRNAs (miR-320a-3p, miR-210-3p, miR-92a-3p, miR-21-5p, and miR-140-3p) were statistically significantly (Wilcoxon’s rank sum *p* < 0.05) elevated in cases compared to matched controls with individual AUCs ranging from 0.57–0.62 (Figure 1A and Table 3; Appendix A). The performance of individual miRNA candidates was associated with stage at time of diagnosis, (Appendix A). Stratification of cases according to lung cancer histological subtype revealed the predictive performance of individual miRNA candidates was conserved (Figure 1B).

We next focused on those miRNAs that were statistically significantly elevated in case sera and, using logistic regression models, developed a combination rule. The combination of miR-21-5p + miR-320a-3p + miR-210-3p was identified as the highest performing panel with a resultant AUC of 0.63 (95% CI: 0.56–0.69) and 22.6% sensitivity at 95% specificity (Figure 2).

Using logistic regression models, we further determined whether the combination of miRNAs plus our previously validated 4MP would yield improved classification performance in differentiating lung cancer cases from controls when compared to the 4MP alone. A combined panel of the three-marker miRNA panel plus the 4MP yielded an AUC of 0.81 (95% CI: 0.73–0.89) (Figure 3A,B). In comparison to the 4MP alone, the combined miRNA panel plus 4MP resulted in statistically significantly improved sensitivity (19.1% (95% CI of difference 0.0–28.6); 2-sided *p*: 0.006) at 95% specificity (Table 4).

## 4. Discussion

Earlier interception of lung cancer is directly linked with a more favorable outcome. Blood-based lung cancer biomarkers offer utility for assessment of individual risk, detection of asymptomatic disease and stratification of equivocal findings from PET or CT imaging. Such blood-based assays would integrate directly into existing clinical workflows and are especially ideal for implementation in a primary care setting. Here, we evaluated 30 miRNA candidates and identified a marker panel comprising three miRNAs for identifying individuals at high risk of developing or harboring disease. Importantly, the combination of miRNAs with a previously validated 4MP protein panel yielded significantly improved sensitivity at the highest specificity thresholds. When applied in a screening context, such performance would allow for identification of additional cases with a minimal false-positive burden.

We note some further considerations with respect to this study. Serum specimens were sourced from Carotene and Retinol Efficacy Trial (CARET) cohort samples for which enrolment was restricted to individuals at high-risk for developing lung cancer, based on smoking-history and/or asbestos exposure and age between 45 and 69 years. Accordingly, the predictive performance of the combined model outside of this age range or for lower-risk populations could not be determined. The performance of the combined miRNA plus 4MP among different ethnic and racial groups was likewise not assessed. Our study did not include an external cohort for further validation of the combined biomarker panels.

Indeterminate lung nodules identified by imaging present a challenging diagnostic scenario. Both the 4MP protein panel and miRNA candidates have been previously shown to contribute towards assessing indeterminate pulmonary nodules and offer improved performance compared to nodule size alone in predicting likelihood of cancer, supporting the observed association between the miRNA plus 4MP panel markers and lung cancer [10,15]. Furthermore, miRNA is known to regulate protein expression and mechanistic link between dysregulated miRNA and proteins in the context of cancer pathophysiology has been previously established [140,141,142,143,144].

## 5. Conclusions

A panel of miRNA plus an existing four-protein marker panel yielded notable improvements in sensitivity at high specificity compared to the protein panel alone. Our findings demonstrate the merits of including miRNAs when developing biomarker panels for lung cancer detection or risk assessment and suggest important implications for improving lung cancer screening and detection that warrant additional study.

## Figures and Tables

**Figure 1 cancers-14-04221-f001:**
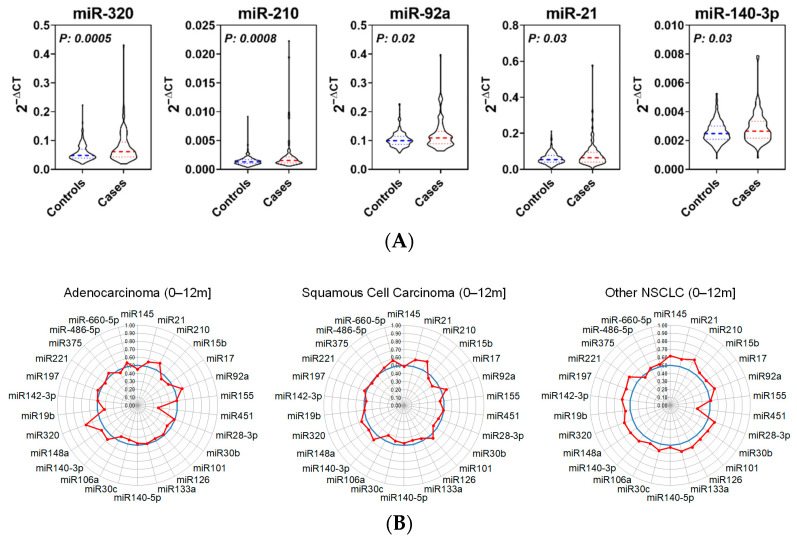
Expression and performance of panel miRNAs in lung cancer validation cohort. (**A**) Relative expression of top-performing upregulated miRNAs in prediagnostic lung cancer case sera (*N* = 102) and matched controls (*N* = 212). (**B**) Radar charts indicating predictive performance of panel miRNAs stratified according to lung cancer subtype. NSCLC, Non-Small-Cell Lung Carcinoma.

**Figure 2 cancers-14-04221-f002:**
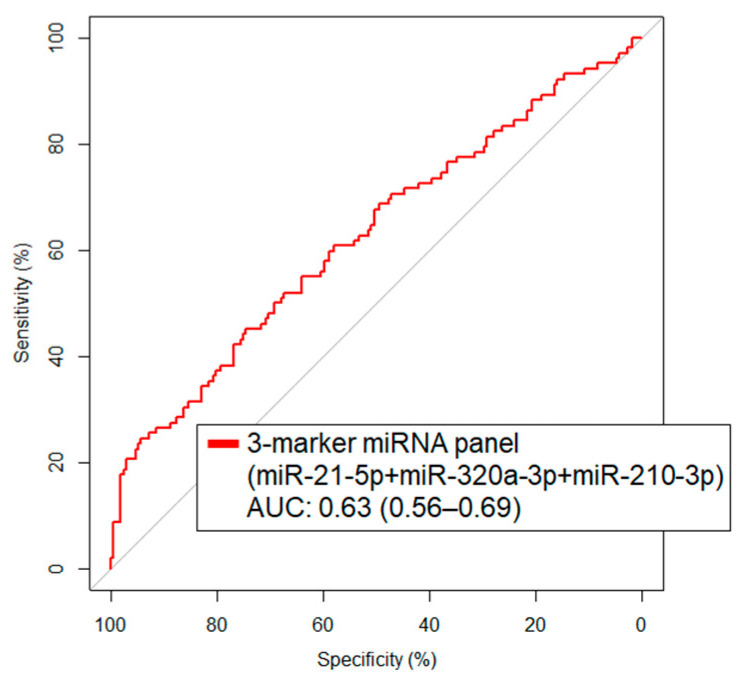
Predictive performance as indicated by area under the curve (AUC) for 3-marker miRNA panel miRNAs in lung cancer prediagnostic serum validation cohort.

**Figure 3 cancers-14-04221-f003:**
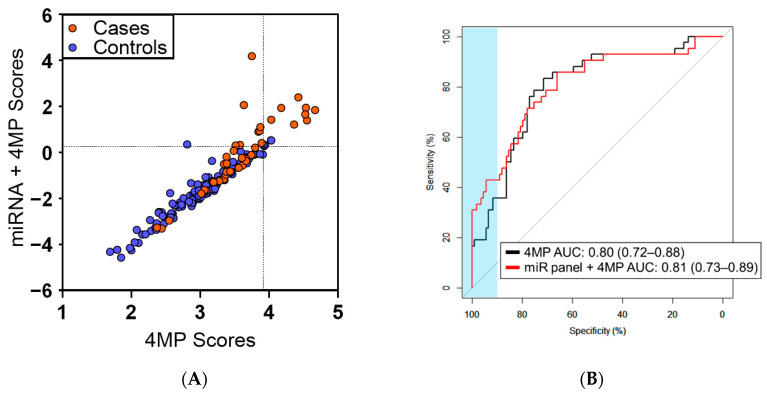
Predictive performance of miRNA panel for distinguishing lung cancer cases diagnosed within one year of blood draw from controls. (**A**) Scatter plot comparing combined three-marker miRNA plus protein 4MP panel scores (y-axis) and the 4MP scores (x-axis) among cases diagnosed within one year of blood draw (orange nodes) and controls (blue nodes). Dashed lines indicate 95% Specificity thresholds. (**B**) ROC curves illustrating discrimination performance of the 3-marker miRNA plus protein 4MP panel and the 4MP alone.

**Table 1 cancers-14-04221-t001:** Serum cohort patient and tumor characteristics. StDev, Standard Deviation.

Variable	Cases	Controls
*N*	102	212
Age, mean ± StDev	65 ± 6	65 ± 6
Sex, *N* (%)		
Female	32 (31%)	63 (30%)
Male	70 (69%)	149 (70%)
Smoking Status, *N* (%)		
Current	70 (69%)	143 (67%)
Former	32 (31%)	69 (33%)
Smoking Pack Years (PYs), mean ± StDev	53.5 ± 21.4	48.7 ± 20.5
Histology, *N* (%)		
Adenocarcinoma	37 (36%)	-
Squamous Cell Carcinoma	27 (26%)	-
Other Non-Small-Cell Lung Carcinoma	38 (37%)	-

**Table 2 cancers-14-04221-t002:** miRNA candidates for serum-based early detection of lung cancer.

Hsa-miRNA	Alternate ID(s)	Validated Reports of Dysregulation in Lung Cancer Clinical Specimens		TaqMan Assay ID	References
		Tissue	Biofluid		
miR-15b-5p	miR-15b		3	390	[9,10,16]
miR-17-5p	miR-17	2	11	2308	[9,10,11,16,17,18,19,20,21,22,23,24]
miR-19b-3p	miR-19b; miR-19b-1	1	9	396	[9,10,22,25,26,27,28,29,30,31]
miR-21-5p	miR-21	7	45	397	[9,10,18,23,25,28,29,32,33,34,35,36,37,38,39,40,41,42,43,44,45,46,47,48,49,50,51,52,53,54,55,56,57,58,59,60,61,62,63,64,65,66,67,68,69,70,71,72,73,74,75,76]
miR-28-3p			2	2446	[9,10]
miR-30b-5p	miR-30b		5	602	[9,10,11,12,77]
miR-30c-5p			5	419	[9,10,11,12,78]
miR-31-5p	miR-31	5	6	2279	[25,32,34,58,68,70,79,80,81,82,83]
miR-92a-3p	miR-92a-2		6	431	[9,10,11,12,16,84]
miR-101-3p	miR-101	1	1	2253	[10,85]
miR-106a-5p	miR-106a		4	2169	[9,10,53,86]
miR-126-3p	miR-126	6	21	2228	[9,10,11,25,28,32,34,41,51,55,68,70,71,74,87,88,89,90,91,92,93,94,95,96,97,98,99]
miR-133a-3p	miR-133a	3	1	2246	[10,100,101,102]
miR-140-3p		2	2	2234	[9,10,79,103]
miR-140-5p			4	1187	[9,10,11,12]
miR-142-3p			3	464	[9,10,11]
miR-145-5p	miR-145	2	13	2278	[10,13,23,32,39,49,51,57,60,71,95,96,98,104,105]
miR-148a-3p	miR-148; miR-148a	2	5	470	[10,11,12,44,63,106,107]
miR-155-5p	miR-155	2	9	2623	[38,40,49,57,61,67,70,108,109,110,111]
miR-182-5p	miR-182	6	10	2334	[17,23,25,34,38,55,61,68,70,71,79,93,112,113,114,115]
miR-197-3p	miR-197		5	497	[9,10,31,38,61]
miR-203a-3p	miR-203	2	1	507	[17,116,117]
miR-205-5p	miR-205	8	15	509	[23,28,32,34,35,51,62,68,70,72,77,80,88,95,96,99,114,116,118,119,120,121,122]
miR-210-3p	miR-210	7	20	512	[9,17,23,34,41,51,54,55,66,67,68,70,79,83,88,89,93,95,96,98,99,107,121,123,124,125,126]
miR-221-3p	miR-221	2	7	524	[9,10,13,23,29,94,120,127,128]
miR-320a-3p	miR-320		3	2277	[9,10,13]
miR-375		2	5	564	[22,68,70,71,129,130,131]
miR-451a	miR-451	5	7	1141	[9,10,25,31,34,35,74,97,132,133,134]
miR-486-5p	miR-486	7	16	1278	[9,10,11,34,35,41,50,52,54,55,68,70,71,75,79,88,108,122,126,134,135,136,137]
miR-660-5p	miR-660		3	1515	[9,10,138]

**Table 3 cancers-14-04221-t003:** Performance of miRNAs for distinguishing lung cancer cases diagnosed within one year following blood draw from matched controls. AUC, Area Under the Receiver Operating Characteristic curve; Ctrl, Control.

	Case:Ctrl		
Hsa-miRNA	Fold-Change	AUC	*p*
miR-451a	0.88	0.37	0.0002
miR-320a-3p	1.36	0.62	0.0005
miR-210-3p	1.63	0.62	0.0008
miR-92a-3p	1.16	0.58	0.0217
mir-21-5p	1.33	0.58	0.0258
miR-140-3p	1.11	0.57	0.0318
miR-148a-3p	1.14	0.56	0.1135
miR-660-5p	1.24	0.55	0.1264
miR-106a-5p	0.97	0.46	0.2274
miR-197-3p	1.23	0.54	0.2274
miR-17-5p	1.01	0.47	0.3960
miR-101-3p	1.05	0.53	0.4178
miR-221-3p	1.04	0.52	0.4789
miR-15b-5p	0.98	0.48	0.4880
miR-30b-5p	1.02	0.48	0.5065
miR-155-5p	1.02	0.48	0.5686
miR-142-3p	1.06	0.52	0.6024
miR-30c-5p	1.02	0.48	0.6238
miR-19b-3p	1	0.49	0.6705
miR-140-5p	0.96	0.49	0.6988
miR-145-5p	1.07	0.51	0.7983
miR-375	1.11	0.51	0.8771
miR-28-3p	1.07	0.5	0.8949
miR-486-5p	1.08	0.5	0.9118
miR-126-3p	1.05	0.5	0.9762
miR-133a-3p	1.1	0.5	0.9815

**Table 4 cancers-14-04221-t004:** Improvement in predictive performance of 4MP panel when combined with 3 miRNAs for distinguishing lung cancer cases from controls in prediagnostic serum validation cohort. CI, Confidence Interval.

Specificity	4MP Sensitivity	4MP + 3 miRNA Sensitivity	Δ (95% CI)	*p*
99%	19.0%	31.0%	11.9% (2.38, 23.9)	0.031
98%	19.0%	33.3%	14.3% (4.8, 26.2)	0.019
95%	19.0%	38.1%	19.1% (0.0, 28.6)	0.006
90%	35.7%	42.9%	7.1% (-4.8, 26.2)	0.369

## Data Availability

Relevant data supporting the findings of this study are available within the Article and Appendix A or are available from the authors upon reasonable request.

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
