# Peer review of "Contributions of Circulating microRNAs for Early Detection of Lung Cancer"

_cancers, 2022, doi:10.3390/cancers14174221_

Round 1

Reviewer 1 Report

In the present study, the authors were interested in the combination of the quantity of circulating miRNAs in association with marker proteins vis-à-vis the early detection of lung cancer.

This work is very interesting, but various biases are present and need to be improved.

- 1: The authors point out in the conclusion that the cohort of individuals chosen for the analysis is not exhaustive of the population. In this sense, they point out the exclusion of people under 45 and over 69, … My remark is that this cohort comes from the CARET trial, which aimed to test the effectiveness of carotene and retinol in cancer prevention. For this, people consumed a combination of 30 mg of beta-carotene and 25,000 IU of retinyl palmitate daily. The sera used in the present study come from people subjected to a particular diet and do not reflect the reality of the population at risk.

- 2: The work carried out is very serious, but why not also take into consideration miRNAs whose expression is reduced, such as miR-198, miR-361-3p, etc.?

- 3: In the article the authors were interested in all the circulating miRNAs. It would have been interesting to distinguish exosomal miRNAs from plasma miRNAs.

- 4 - The prediction results are very promising, but they are based on small quantitative variations of the miRNAs, certainly significant, but weak. Are these variations relevant ?

Although the analyzes are seriously carried out and the results obtained are very relevant, I think that this article is too premature and requires verification. I therefore issue a negative opinion for the publication of this article in cancers.

Reviewer 2 Report

For the early detection of lung cancer, Vykoukal and collogues suggested that the contribution of circulating miRNAs may aid in the early detection of lung cancer and to develop new biomarkers. They have therefore analyzed 30 miRNAs in an individualized and synergistic manner to determine cases of lung cancer. Five of these 30 miRNAs assessed (miR-320a-3p, miR-210-3p, miR-92a-3p, miR-21-5p, and miR-140-3p) were statistically significant. While around 4MP alone, the synergy of 3-miRNAs (miR-21-5p + miR-320a-3p + miR-210-3p) + 4MP improved the sensitivity at 95%. As such, these authors suggested that synergistically these miRNA + 4MP led to significant progress in sensitivity at higher specificity than 4MP alone. Therefore, it has novelty. The manuscript is well-written. Most of the description is clear. The conclusions are consistent with the evidence and arguments presented. In addition, it also provided a comprehensive knowledge to our understanding regarding using a combination of miRNAs early detection of lung cancer. Therefore, this study is very useful for the clinical perspective.

There are two minor points the authors need to revise:

Figure1 A and B: A & B, should be right-hand side.

Figure 1B: Its hard to see/read the miRNAs in each radar graph. Please provide high-resolution figure 1B.

Reviewer 3 Report

The manuscript by Vykoukal et al entitled “Contributions of circulating microRNAs for early detection of  lung cancer” provide evidences for the use of miRNAs in combination with 4MP (4 protein marker panel) for the early detection of lung cancer. The study is in general very well performed, and interesting. Overall, the data emphasize the importance of circulating miRNAs for early detection of lung cancer.

Specific comments:

  • In Figure 1B, the labelling of the radar charts is a bit difficult to read. Please increase the font size of the miRNAs labelling.

  • Based on Table 3, there are six miRNAs that were significant based on the p-value. Why the authors choose only the three of them, miR-21-5p, miR-320a-3p, miR-210-3p? It’s worth doing a predictive performance with all six of them, and compare the difference with the 3-marker miRNA?

  • Do the authors believe that the combination of 6-marker miRNA + 4MP will give a better sensitivity than the 3-marker miRNA + 4MP?

  • The study includes the smoking status (current/former) of the cases and control population. However, there is a proportion of patients with lung cancer that are non-smokers. What miRNAs are upregulated in those patients? Do the authors believe that this combination of miRNAs panel + 4MP will still apply for this category as an early detection method? It is worth explaining this further in the discussion.

Round 2

Reviewer 1 Report

Thanks to the authors for responding to my objections. The manuscript is more readable, especially at the level of the figures.
Good work
Congratulation